# Efficient Removal of Pb(II) from Aqueous Medium Using Chemically Modified Silica Monolith

**DOI:** 10.3390/molecules26226885

**Published:** 2021-11-15

**Authors:** Ashraf Ali, Sarah Alharthi, Bashir Ahmad, Alia Naz, Idrees Khan, Fazal Mabood

**Affiliations:** 1Department of Chemistry, The University of Haripur, Haripur 22620, Pakistan; 2Department of Chemistry, Taif University, Taif 21944, Saudi Arabia; sarah.alharthi@tu.edu.sa; 3Department of Biology, The University of Haripur, Haripur 22620, Pakistan; bashir.ahmad@uoh.edu.pk; 4Department of Environmental Science, The University of Haripur, Haripur 22620, Pakistan; alia.naz@uoh.edu.pk (A.N.); idreeskan@yahoo.com (I.K.); 5Institute of Chemical Sciences, University of Swat, Haripur 19200, Pakistan; fazal@uswat.edu.pk

**Keywords:** heavy metals, adsorption, lead, adsorption capacity, silica monolith particles

## Abstract

The adsorptive removal of lead (II) from aqueous medium was carried out by chemically modified silica monolith particles. Porous silica monolith particles were prepared by the sol-gel method and their surface modification was carried out using trimethoxy silyl propyl urea (TSPU) to prepare inorganic–organic hybrid adsorbent. The resultant adsorbent was evaluated for the removal of lead (Pb) from aqueous medium. The effect of pH, adsorbent dose, metal ion concentration and adsorption time was determined. It was found that the optimum conditions for adsorption of lead (Pb) were pH 5, adsorbent dose of 0.4 g/L, Pb(II) ions concentration of 500 mg/L and adsorption time of 1 h. The adsorbent chemically modified SM was characterized by scanning electron microscopy (SEM), BET/BJH and thermo gravimetric analysis (TGA). The percent adsorption of Pb(II) onto chemically modified silica monolith particles was 98%. An isotherm study showed that the adsorption data of Pb(II) onto chemically modified SM was fully fitted with the Freundlich and Langmuir isotherm models. It was found from kinetic study that the adsorption of Pb(II) followed a pseudo second-order model. Moreover, thermodynamic study suggests that the adsorption of Pb(II) is spontaneous and exothermic. The adsorption capacity of chemically modified SM for Pb(II) ions was 792 mg/g which is quite high as compared to the traditional adsorbents. The adsorbent chemically modified SM was regenerated, used again three times for the adsorption of Pb(II) ions and it was found that the adsorption capacity of the regenerated adsorbent was only dropped by 7%. Due to high adsorption capacity chemically modified silica monolith particles could be used as an effective adsorbent for the removal of heavy metals from wastewater.

## 1. Introduction

Water pollution is a very serious problem for humans, animals and plants in the world. A large number of industries produce pollutants which contaminate air, soil and water. Today, the purification of environmental resources such as air, water and soil is an important issue of concern for scientists and researchers. Water pollution by heavy metals is a major problem for human, animals and plants due to their accumulation in food chains and high toxicity. Lead (Pb) is one of the toxic heavy metals which gets into the blood stream of humans and animals and accumulates in their bones [1,2,3]. Pb is used in various industrial activities such as smelting, mining, printing, petroleum refining, battery manufacturing and pigment production [4,5,6,7]. Several industries utilize lead for a variety of applications, such as electroplating, battery manufacturing, petrochemical processes, fuels, photographic materials, printing pigments and explosive manufacturing. Pb has adverse effects on the immune system, nervous system and reproductive system in humans. Various methods are used for the removal of heavy metals from water such as ion exchange, electrolysis, reverse osmosis, oxidation reduction, chemical precipitation, and solvent extraction. These methods have some limitations, such as low metal removal capacity (particularly for wastewater treatment containing very low concentrations of heavy metals [8,9]), the consumption of high energy and secondary waste production, high cost [10,11], the production of toxic sludge [12,13] and complicated operation procedures. An ideal water treatment technique should be environmentally sustainable; one that does not create any pollution and has a low cost.

Adsorption is one of the best techniques used for the removal of heavy metals from wastewater. Two types of adsorbents, noncompetitive and competitive adsorbents, are used for the removal of heavy metal ions. For only one kind of metal ions, noncompetitive adsorbents are used, while for the removal of two or more metal ions from wastewater, competitive adsorbents are used. Non-competitive adsorbents remove twice the amount of metal ions than the competitive adsorbents. During competitive adsorption each adsorbent adsorbs metal ions simultaneously, depending upon the affinity of metal ions for a particular functional group of adsorbent. Activated carbon is used as a good adsorbent for the remediation of polluted water but, due to its high-cost, agricultural products and by-products are used as cost-effective adsorbents for the removal of metal ions from aqueous stream.

Various adsorbents are used for the removal of Pb(II) ions from water, such as natural biosorbents [14,15,16,17], chemically treated agricultural products and byproducts [14,18,19,20,21,22], composites [23,24,25,26,27,28], silica particles [29] and several other adsorbents [30,31,32,33,34,35,36,37]. These adsorbents have several limitations such as lower adsorption capacities, poor interaction with metal ions in aqueous medium, difficult multistep preparation methods, the production of additional wastes during preparation and their poor regeneration which limits their use. An ideal adsorbent for the removal of heavy metals ions should have a low cost, high affinity for metal ions, high adsorption capacity, and easy regeneration. Several adsorbents, such as activated carbon, have a higher adsorption capacity but due to its high cost it cannot be applied on a large scale for wastewater treatment. On the other hand, various low-cost adsorbents based on agricultural waste, fruits peels, cellulose, etc., have a low cost but their adsorption capacity and separation efficiency are poor. So, there is always a demand for the preparation of an efficient low-cost adsorbent which can effectively remove toxic heavy metals and organic pollutants from industrial wastewater [38,39,40,41,42,43,44].

In the current study, a novel adsorbent was prepared based on silica monolith particles. Silica particles were selected owing to their large surface area, porous structure composed of macro-pores, mesopores and micro-pores and easy surface modification to make inorganic-organic hybrid adsorbents. Firstly, porous silica monolith particles were prepared by the sol-gel method [45]. Then, these particles were chemically modified with trimethoxy silyl propyl urea (TSPU) to prepare inorganic–organic hybrid adsorbent for the removal of Pb ions. The adsorbent chemically modified SM was characterized by SEM, and TGA. The adsorption of Pb(II) from synthetic solution and industrial wastewater was carried out using chemically modified SM. The effects of metal ion concentration, adsorbent dose, adsorption time and pH were determined during batch adsorption experiments. The optimum conditions for adsorption of Pb(II) were pH 5, an adsorbent dose of 0.4 g/L, a metal concentration of 500 mg/L and an adsorption time of 45 min. It was found that the percent adsorption of Pb(II) onto chemically modified SM was 96%. An isotherm study shows that the adsorption of Pb(II) ions onto chemically modified SM was in close agreement with Freundlich and Langmuir isotherm models. Kinetic study show that the adsorption of Pb(II) follow pseudo second-order reaction, while the thermodynamic study shows that the adsorption of Pb(II) onto chemically modified SM is exothermic and spontaneous. Owing to a high adsorption capacity, chemically modified SM could be used as an effective adsorbent for Pb(II) and other heavy metals removal from wastewater. The developed adsorbent could be an alternative to expansive activated carbon.

## 2. Materials and Methods

### 2.1. Chemicals and Apparatus

PEG (polyethylene glycol), urea, acetic acid, trimethoxy silyl propyl urea, methanol, acetone, nitric acid, ethylene diamine tetra acetate and lead nitrate were bought from Sigma-Aldrich (St. Louis, MO, USA). All chemicals were used as such without further purification.

### 2.2. Preparation of Silica Monolith Particles

Silica monolith particles were prepared according to our developed protocol [45]. In a typical experiment PEG (8 mg) and urea (8.5 mg) were dissolved in 0.01 N acetic acid (10 mL) and stirred for 10 min. TMOS (2.4 mL) was added into the mixture and stirred for 10 min under ice-cold conditions. The reaction mixture was heated at 40 °C for 5 h followed by heating in an autoclave for 5 h at 120 °C. Silica particles were dried and calcined at 500 °C for 3 h in a muffle furnace.

### 2.3. Chemical Modification of Silica Monolith Particles

Dried silica monolith particles (1 g) were suspended in 10 mL anhydrous toluene and stirred for 10 min. Then, 1-[3-(trimethoxysilyl)propyl]urea (0.1 mL) was dissolved in 1 mL toluene and added to the reaction flask under reflux for 5 h. The product was washed filtered and dried at 70 °C for 5 h.

### 2.4. Characterization of Adsorbent

The pore volume and surface area of the adsorbent was characterized by N_2_ adsorption/desorption isotherms while surface morphology of adsorbent was characterized by S-4200 field emission scanning electron microscopy.

### 2.5. Adsorption of Pb(II) on Chemically Modified Silica Monolith

The adsorption of Pb(II) was carried out in a batch system. The stock solution of lead nitrate (1000 mg L^−1^) was diluted to 500 mg L^−1^ and 500 mL was added into a 1-L conical flask containing 100 mg of the adsorbent (chemically modified SM). The mixture was agitated at 100 rpm in a water bath at 25 °C for 1 h. The mixture was filtered after and the Pb(II) ions concentration in the filtrate was determined by atomic absorption spectrometer (Perkin Elmer A. Analyst 800 (Platinum Elmer Enterprise Management Co., Ltd. Massachusetts, MA, USA)). The amount of Pb(II) ions adsorbed was determined from the difference between the concentrations of the Pb(II) solutions before and after adsorption. The same procedure was applied for the removal of Pb(II) ions from industrial wastewater. The amount of Pb(II) adsorbed per gram of adsorbent was calculated by the following Equation (1)
(1)$$qe=\frac{V\left(C_{0}-C_{e}\right)}{W}$$
where *qe* is the quantity of Pb(II) adsorbed, *V* is the volume of solution (L), *W* is the amount of adsorbent (g), and *C*_0_ and *C_e_* are the initial and equilibrium concentration of Pb(II) ions in solution, respectively.

The percent Pb(II) removal was calculated using the following Equation (2)
(2)$$\mathrm{Pb}\left(\mathrm{II}\right)\;\mathrm{removal}\;\left(\%\right)=\frac{\left(C_{0}-C_{e}\right)}{C_{0}}\times 100\;$$

### 2.6. Adsorption Isotherm Study

The adsorption equilibrium data of Pb(II) on chemically modified SM were computed by Freundlich, Langmuir and Temkin isotherm models. The regression coefficient values were judged to find the applicability of these models for the adsorption of Pb(II) ions onto chemically modified SM. Microsoft Excel and Origin Pro-8 were used for linear regression calculations.

### 2.7. Kinetic Study

Lead nitrate solution (500 mg L^−1^) was taken in a flask with 0.4 g adsorbent chemically modified SM, stirred for different times, filtered and the Pb(II) concentration in the filtrate was analyzed by atomic absorption spectrometer. Optimum adsorption time was determined, and this optimized time was used for further analysis.

### 2.8. Batch to Batch Reproducibility

Three batches of chemically modified SM were prepared and adsorption of Pb(II) from the aqueous solution was carried out under optimized conditions on each batch. It was found that the adsorption capacity of each batch of chemically modified SM was very close. Similarly, each experiment was repeated three times and the values of standard deviations were calculated.

### 2.9. Experimental Quality Evaluation

All adsorption experiments were carried out in triplicates to ensure the reproducibility of results. The relative standard deviation was used as the error parameter for all analysis and the value for each set of measurements was <1.7%. Each experimental set was carried out using blanks to ensure the elimination of errors associated with experimental conditions. For each experimental analysis procedure, blanks were prepared using distilled water and the blank samples were subjected to the same treatment process using the same type of experimental vessel. In the analysis of Pb(II) ions in solution, the blank samples were also analyzed first prior to analysis of the standards and the samples. A calibration curve for each set of measurements was prepared using the standards prepared for the Pb(II) ions.

A derivative of Marquardt’s percent standard deviation (MPSD) was examined as an error function and a set of isotherm parameters was determined by minimizing the respective error function across the concentration range studied. The MPSD was calculated using Equation (3).
(3)$$\overset{N}{\underset{i=1}{\sum }}{\left[\frac{q_{e\left(exp\right)}-q_{e\left(cal\right)}}{q_{e\left(exp\right)}}\right]}_{i}^{2}$$

## 3. Results and Discussions

### 3.1. Morphology of the Adsorbent

The scanning electron microscopy images of bare silica monolith and chemically modified silica monolith are shown in Figure 1A,B, respectively. Bare silica monolith (Figure 1A) looks rougher than chemical modified silica monolith because after chemical modification a thin layer of methoxy silyl propyl urea develops on the silica monolith. Figure 1C,D shows the SEM images before and after the adsorption of lead (II) ions, respectively. The surface of Pb(II) adsorbed chemically modified silica monolith (Figure 1D) looks rougher due to the attachment of Pb(II) ions than chemically modified silica monolith before the adsorption of Pb(II) ions.

### 3.2. Particle Size and Pore Size Distribution of the Adsorbent

The volume-based particle size distribution of bare and chemically modified silica monolith is given in Figure 2A. The particle size data corresponding to particular fractions d(0.5) for volume-based particle size distribution are summarized in Table 1. The average particle sizes of bare and chemically bound silica monolith particles are 2.4 and 2.8 µm, respectively. The results shows that particle size increased by 0.4 µm after chemical modification with trimethoxy silyl propyl urea. The BET (Brunauer–Emmett–Teller) nitrogen adsorption/desorption analysis data of bare and chemically modified silica monolith are summarized in Table 1. The pore size distributions of bare and chemically modified silica monolith are presented in Figure 2B. The BET surface area of bare silica monolith and chemically modifies silica monolith was 124 m^2^/g and 113 m^2^/g while the average pore size of bare and chemically modified silica monolith was 342 and 310 Å. respectively. The results in Table 1 and Figure 1B shows that pore size, pore volume and surface area of silica monolith decreased after chemical modification.

### 3.3. The Effect of Concentration, Adsorbent Dose, Contact Time and pH on Pb(II) Removal

#### 3.3.1. The Effect of Concentration

The effect of concentration on Pb(II) removal by chemically modified SM was studied by changing the metal ion concentration in the solution and keeping other parameters constant (adsorbent dose (0.4), pH at 5.5, temperature at 25 °C). Lead nitrate solutions of 100, 200, 300, 500, 600, 700 and 800 mg/L were prepared and put into conical flasks separately, containing 0.4 g adsorbent and shaken at 25 °C for 1 h in a mechanical shaker. After the specified time, the contents of each flask was filtered and the concentration of Pb(II) ions was determined in each filtrate by atomic absorption spectrometer. The results show that increasing the Pb(II) concentration decreases its removal. It may be owing to the occupation of active sites of the adsorbent by metal ions. A further increase in metal ion concentration produces more ions and at a specified time the adsorbent sites cannot pick more ions from the solution. The curve in Figure 3A shows that increasing Pb(II) concentration decreases the %Pb(II) removal. At a low metal ion concentration (100 mg/L) the adsorption capacity of adsorbent is at maximum at 94%, and the % removal of Pb(II) ions decreased with increasing concentration. The data presented in Figure 3A show that the developed adsorbent (chemically modified SM) have good adsorption capacity for Pb(II) ions even at higher metal ions concentration. The Pb(II) removal at 400 mg/L and 600 mg/L is 75% and 60%, respectively. It is expected that the adsorbent could be used for highly contaminated waste water with Pb(II) ions.

#### 3.3.2. The Effect of Adsorbent Dose

The curve in Figure 3B shows that removal of Pb(II) ions increases with the increase in the adsorbent dose until the optimum dose (0.4 g/L). Initially, when a less amount of adsorbent (1 mg/L) was used for the removal of Pb(II) ions from 400 mg/L of lead nitrate solution, only 40% of Pb(II) ions were removed because the amount of adsorbent was not enough for such high concentration of Pb(II) solution (400 mg/L). As the amount of adsorbent was increased in the second flask to 0.2 g/L, about 65% of Pb(II) ions were removed, as shown the Figure 3B. Similarly the same trend was observed until the adsorbent amount was increased to 0.4 g/L, where the Pb(II) ion removal was at maximum (96%). When the adsorbent amount was increased beyond 0.4 g/L to 0.6 g/L, there was no increase in the removal efficiency and the same 96% removal of Pb(II) was recorded, as shown in the Figure 3B. These results show that 0.4 g/L of adsorbent have enough active sites which can adsorb Pb(II) ions 400 mg/L Pb(II) solution. Another interesting finding of this study is the higher adsorption capacity at a low dose of the current adsorbent (0.4 g/L). In the literature, the adsorbent dose is usually used in grams rather than milligrams, while in this study the adsorbent is based on silica monolith particles which are highly porous, and the ligand attached have higher affinity for metal ions.

#### 3.3.3. The Effect of Contact Time

The effect of adsorption time on Pb(II) adsorption by chemically modified SM is shown in Figure 3C. For studying the effect of contact time, other parameters were kept constant, such as the temperature at 25 °C, pH at 5, the agitation speed at 300 rpm, the lead ion concentration at 400 mg/L and the adsorbent dose at 0.4 g/L. Figure 3C shows that Pb(II) removal increases with increasing adsorption time until equilibrium was established in 1 h and the % removal reached maximum at 96%. A further increase in adsorption time beyond 1 h has no effect and the removal % is same as 96%. Initially the adsorption was increased with increasing adsorption time because the metal ions interact with adsorbent active sites and thus their removal increased until the establishment of equilibrium in 1 h. After 1 h, there was no change in metal ion removal because the active sites of the adsorbent were already saturated with Pb(II) ions.

#### 3.3.4. The Effect of pH

The effect of pH on Pb(II) removal by chemically modified SM was investigated by keeping temperature, adsorption time, agitation speed, Pb(II) ion concentration and the adsorbent dose constant. The effect of pH on Pb(II) removal is shown in Figure 3D. The results show that the Pb(II) removal is lower at very low pH, and increased with increasing pH up to 5. At lower pH the carbonyl groups of the adsorbent are protonated and H^+^ ions act as competitor for positively charged Pb(II) ions. Thus, the interaction between Pb(II) ions and adsorbent decreases. When pH increases beyond 5, the removal of Pb(II) decreases due to the competition of nitrate ions and OH^-^ ions where the letter is a more prominent species. At higher pH, the Pb(II) removal decreases due to the formation of soluble hydroxyl complexes of lead.

### 3.4. Adsorption Isotherm Study

An adsorption isotherm equation is an expression of the relation between the amount of solute adsorbed and the concentration of the solute in the liquid phase. Freundlich and Langmuir adsorption models were used to study the interaction of Pb(II) with adsorbent.

#### 3.4.1. Freundlich Adsorption Isotherm

According to Freundlich isotherm the adsorption is multilayer while adsorption takes place on heterogeneous surface and increases with the increase in concentration until equilibrium get established. The same pattern was observed for the adsorption of Pb(II) onto chemically modified SM. The linear form of Freundlich adsorption isotherm is given as [46]
(4)$$logqe=logKf+\frac{1}{n}\;logC_{e}$$

By plotting *logqe* vs. *logC_e_* linear plots were obtained with slope 1/*n*, as shown in Figure 4B. The *n* and *K* (L mg^−1^) (adsorption capacity) were calculated from the graph. The “*n*” values show that the surface of the adsorbent has great affinity for Pb(II) ions [47].

#### 3.4.2. Langmuir Adsorption Isotherm

Langmuir adsorption isotherm can be written as [48],
(5)$$\frac{C_{e}}{q}=\frac{1}{q_{max}k}+\frac{C_{e}}{q_{max}}$$
where *C_e_* (mgL^−1^) is the concentration of Pb(II) ions at equilibrium, *qe* (mgg^−1^) is the amount of Pb(II) ions adsorbed per unit mass of adsorbent, *q_max_* (mgg^−1^) is the maximum adsorption capacity, and *KL* (L mg^−1^) is Langmuir constant [48].

From the data of Pb(II) adsorption, *C_e_*/*q* against *C_e_* was plotted which gave a straight line, as shown in Figure 4A. The equilibrium parameter (*R_L_*) was calculated from Equation (5)
(6)$$R_{L}=\frac{1}{1+K_{L}C_{0}}$$

The isotherm constants and regression values (Table 2) show that the adsorption data of Pb(II) onto chemically modified SM are in close agreement with the Langmuir isotherm model. The results show that the adsorption data is fully fitted with the Langmuir isotherm model (as shown in Figure 4A), the adsorption of Pb(II) onto chemically modified SM is favorable and the R_L_ values are less than 1 (Table 2).

#### 3.4.3. Temkin Adsorption Isotherm

Temkin isotherm explains adsorbent–adsorbate interactions [49]. This model is a useful tool to estimate the adsorption heat, which can be calculated using the following equation
(7)$$qe=\frac{R_{T}}{B_{T}}\;\left(lnK_{T}C_{e}\right)=B_{T}\left(lnK_{T}C_{e}\right)$$
where, $B_{T}=R_{T}/b_{T}$ which is related to the adsorption heat, R is the gas constant (8.314 J/mol K), T (K) is the absolute temperature in Kelvin, $B_{T}$ (J/mol) is the Temkin isotherm constant, which is the variation of adsorption energy and is the equilibrium binding constant corresponding to the maximum binding energy. Both $B_{T}$ and $K_{T}$ can be calculated from the slope and the intercept of the linear plot based on   qe  versus lnCe, respectively.

This model assumes that the heat of adsorption of all molecules in the layer would decrease linearly rather than logarithmically with coverage [49]. The quantity adsorbed *qe* was plotted against *lnCe* and the constants were determined from the slope and intercept Figure 4C. The increase in maximum binding energy (AT) for chemically modified SM with respect to the increase in temperature implies that the system was influenced by thermal properties. The interaction of Pb(II) ions with the adsorption sites was seen more at higher temperatures. The Temkin isotherm (Figure 4C) of chemically modified SM (Table 2) shows that the heat of adsorption increases with increase in temperature, indicating that adsorption of Pb(II) onto chemically modified SM is endothermic. Additionally, the B_T_ values for chemically modified SM is 48.53 kJ mol^−1^, which is quite high, suggesting a strong interaction between Pb ions with chemically modified SM. These results indicate that the process would be physio-sorption [50]. The obtained correlation coefficient for chemically modified SM was high which confirms the better fit of the Temkin model to the experimental data.

### 3.5. Adsorption Kinetics

To find the mechanism and rate determining step of adsorption of Pb(II) onto chemically modified SM, experiments were carried out to find the required time for Pb(II) adsorption. It was found that the Pb(II) adsorption onto chemically modified SM was quick until 1 h and then slowed down until the establishment of the equilibrium. The results show that the Pb(II) ions chemically adsorb onto the adsorbent (chemically modified SM) and interact with its functional groups (carbonyl and amino). The rate determining step was find out using pseudo first-order, pseudo second-order, and intra particle diffusion models.

#### 3.5.1. Pseudo First-Order Kinetic Model

Lagergren pseudo first-order rate expression can be written as in linear form (8),
(8)$$In\left(q_{e}-q_{t}\right)=Inq_{e}-k_{1}t$$

In this equation, the amount of Pb(II) adsorbed (mg/g) on the adsorbent at equilibrium is represented by *qe*, while *qt* represents the adsorption of Pb(II) ions at a specific time (t), and K_1_ is the rate constant of pseudo first-order adsorption per minute. In Figure 5A, *ln*(*qe* − *qt*) vs. *t* plot at various concentrations gave a straight line. The rate constant *k*_1_ (min^−1^) was calculated from the slope of the linear plots. The experimental and calculated values of various parameters such as *qe* (mg/g), *K*_1_, *R*^2^ and *K*_i_ (mgg^−1^min^−1^) for pseudo first-order kinetic model, pseudo second-order kinetic model and intra particles diffusion model are given in Table 3.

#### 3.5.2. Pseudo Second-Order Model

The linear form of the pseudo second-order model may be described as below,
(9)$$\frac{t}{q^{t}}=\frac{1}{K_{2}q_{e}^{2}}+\frac{t}{q_{e}}$$
where *k*_2_ (g·mg^−1^·min^−1^) is the rate constant of the pseudo second-order kinetic equation, *qe* and *qt* are the amount of Pb(II) adsorbed (mg/g) onto chemically modified SM at equilibrium and specific time (t), respectively. The adsorption capacity (*qe*) and rate constant *k*_2_ (g·mg^−1^·min^−1^) were calculated from the linear plot of (*t*) vs. (*t*/*qt*), Figure 5B [51]. The results show that the second-order model is more suitable than the first-order model to describe the adsorption kinetics of Pb(II) onto chemically modified SM.

#### 3.5.3. Intra-Particles Diffusion Model

To investigate the diffusion mechanism of Pb(II) onto chemically modified SM, the intra-particle diffusion model was used [52]. The Pb(II) ions from solution were transferred into the solid phase [53]. The equation can be written as
(10)$$q_{t}=K_{p}t^{0.5}+C$$
where *qt* (mg·g^−1^) is the concentration of Pb(II) adsorbed onto the adsorbent at time *t*, *C* is the intercept and *K_p_* (mg·g^−1^.min^0.5^) is the intra-particle diffusion rate constant. The value of the intercept gives an idea about the boundary layer thickness, i.e., the larger the intercept, the greater the boundary layer effect [51]. It is seen from Table 3 that the value of intercept is high and it increases in the case of the combination of metal ions. This result shows that boundary layer diffusion is the rate controlling step for the adsorption process and it is dominant when the Pb(II) ion concentration is higher.

These models are applicable to the adsorption of Pb(II) and the plots are linear. The *r*^2^ for the pseudo second-order kinetic model is 0.99 and the calculated *qe* values are nearly close to the experimental *qe* values as shown in Table 3. Figure 5C shows that the intra-particle diffusion rate increases with an increase in Pb(II) concentration.

A better correlation was confirmed by the pseudo second-order model to the experimental data followed by intra-particle diffusion for Pb(II) adsorption on the surface of chemically modified SM being identified with high linear regression (*R*^2^) values. It suggests the existence of a certain boundary layer effect on the surface of chemically modified SM. The rate-limiting step involved the chemical adsorption for Pb(II) and chemically modified SM. The calculated *qe* values were more than the experimental obtained values which additionally specify the acceptable desirable quality of the kinetic theory for the adsorption of Pb(II) onto chemically modified SM.

### 3.6. Desorption, Regeneration and Reuse of the Adsorbent

The regeneration of chemically modified SM was carried out and the adsorption capacity of the regenerated adsorbent was investigated after each run. Used adsorbent containing Pb(II) ions was stirred with 50 mL of 0.2 mol/L HNO_3_ and 20 mL of 0.2 mol/L EDTA separately. The adsorbent was stirred at room temperature in 50 mL of 0.2 mol/L NaOH and 20 mL of 0.2 mol/L EDTA, respectively, for 12 h and the filtrate containing desorbed metal ions were separated by centrifugation and filtration. The concentration of Pb(II) ions was determined in the filtrate by atomic absorption spectrometer. The samples were washed with deionized water after each regeneration cycle. The results shows that when chemically modified SM was initially used at concentration 100 mg/L, the loading capacity for Pb(II) ions was 62.66 mg/g, and after the first regeneration with stripping solution, the loading capacity decreased to 46.24 mg/g. The second loading capacity was 37.74 mg/g and the third loading capacity was 34.45 mg/g. It was observed that the adsorption capacity of chemically modified SM was decreased slightly with each of the repeated three regeneration cycles. This decrease adsorption capacity of chemically modified SM may occur due to the inaccessibility of metal ions to the incoming ions from the stripping solution, and thus the active sites of adsorbent reduce with time. The second possible reason may be due to the removal of some active functional groups by the stripping solution and thus the adsorption capacity of the adsorbent decreased.

### 3.7. Comparison of Chemically Modified SM for Pb(II) Removal with Other Adsorbents

The adsorption capacity of chemically modified SM adsorbent prepared in this study for Pb(II) removal is compared with other synthesized adsorbents reported in the literature, as presented in Table 4. Among the reported adsorbents CS/Fe-hydroxyapatite composite bead have the highest adsorption capacity (1385 mg/g) [51]. The adsorption capacity of chemically modified SM for Pb(II) is 792 mg/g.

## 4. Conclusions

In this study, novel adsorbent methoxy silyl propyl urea bonded silica monolith particles were prepared by a two-step synthesis. In step one, fully porous silica monolith particles (3 µm) were prepared by modified sol-gel method. The reaction formulation was adjusted to prepare particles of the desired pore size, particle size and surface area. In the second step, these particles were chemically modified with trimethoxy silyl proyl urea to prepare an adsorbent for the removal of Pb(II) from aqueous medium. Characterization of the developed adsorbent shows that chemically modified SM was successfully prepared. The results show that chemically modified SM can effectively remove Pb(II) from synthetic solution and industrial wastewater contaminated with Pb(II) ions. The results show that chemically modified SM is very effective for the removal of Pb(II) ions. Very little amount of adsorbent can decontaminate Pb(II) solution of high concentration of Pb(II) ions. The higher adsorption capacity of chemically modified SM may be owing to the highly porous nature of silica monolith particles and special functional group (carbonyl and amine). The isotherm study shows that the adsorption data are fully fitted with the Langmuir isotherm model. The correlation coefficient of Langmuir adsorption model was higher than that of the Freundlich adsorption model, which shows that the adsorption is a monolayer on the surface of chemically modified SM adsorbent. The kinetic study shows that the adsorption process can be well described by the pseudo second-order kinetic model, suggesting that main rate-controlling step was chemisorption. The preparation of chemically modified SM adsorbent is easy and robust while its adsorption capacity is quite high (792 mg/g) for Pb(II) as compared to the traditional adsorbents reported in the literature. The *qe* of chemically modified SM for Pb(II) removal is the second highest (792 mg/L) after CS/Fe-hydroxyapatite composite bead (1385 mg/L). The developed adsorbent will be evaluated for the removal of other toxic heavy metal ions and dyes in the future.

## Figures and Tables

**Figure 1 molecules-26-06885-f001:**
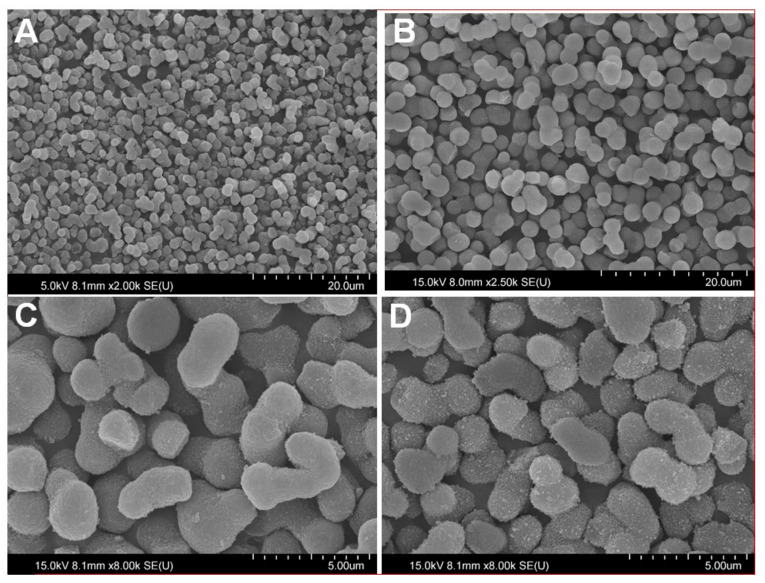
SEM images of bare SM (**A**), chemically modified SM (**B**), broader view of chemically modified SM before adsorption (**C**) and after the adsorption of Pb(II) (**D**).

**Figure 2 molecules-26-06885-f002:**
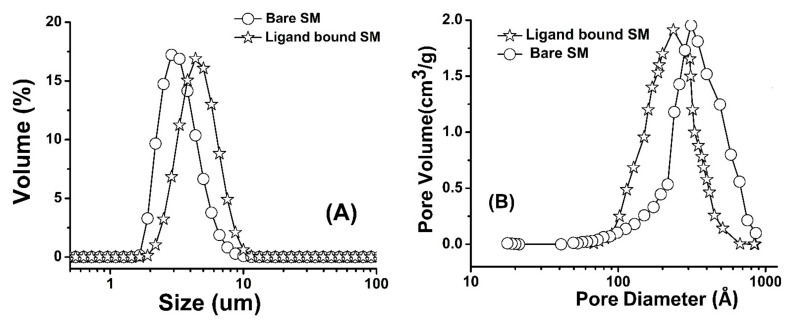
Particle size distribution (**A**) and pores size distribution (**B**) of bare and chemically modified silica monolith.

**Figure 3 molecules-26-06885-f003:**
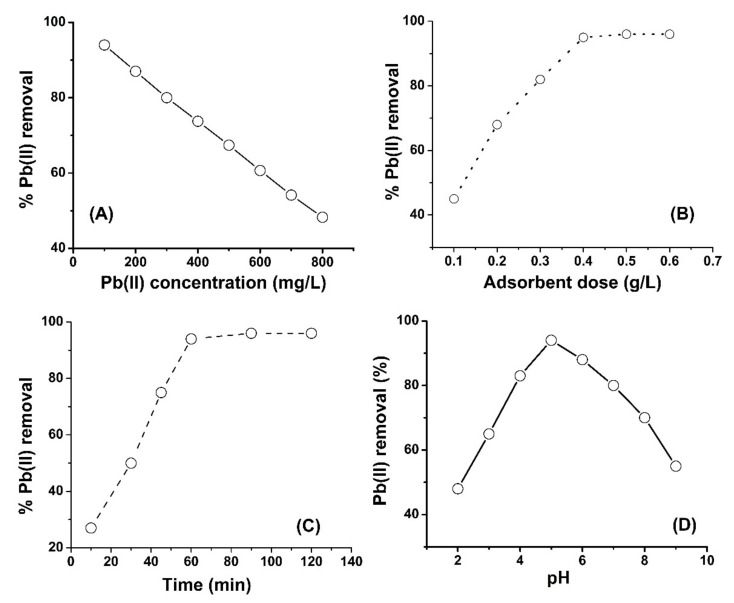
Effect of concentration (**A**), adsorbent dose (**B**), contact time (**C**) and pH (**D**) on Pb(II) adsorption onto chemically modified silica monolith.

**Figure 4 molecules-26-06885-f004:**
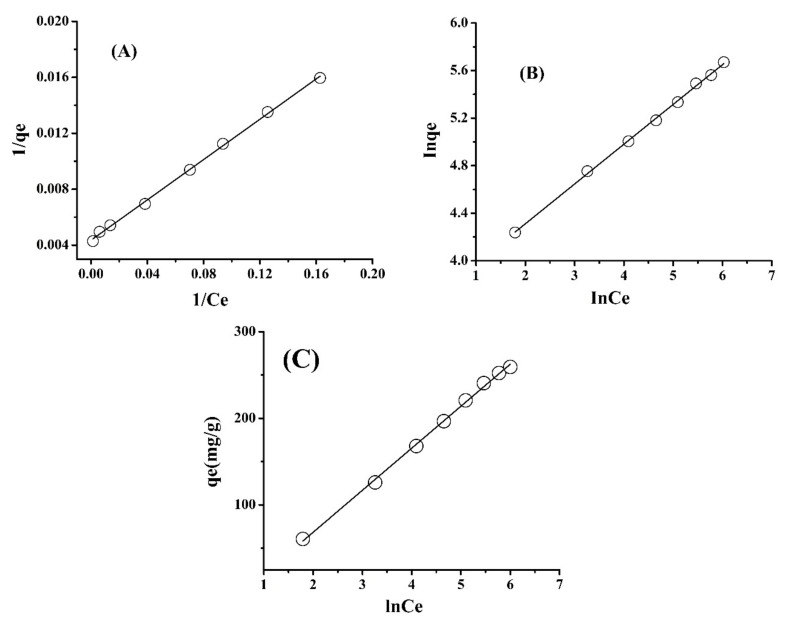
Adsorption isotherm models for Pb(II) adsorption onto chemically modified SM, Langmuir model (**A**) and Freundlich model (**B**) and Temkin (**C**).

**Figure 5 molecules-26-06885-f005:**
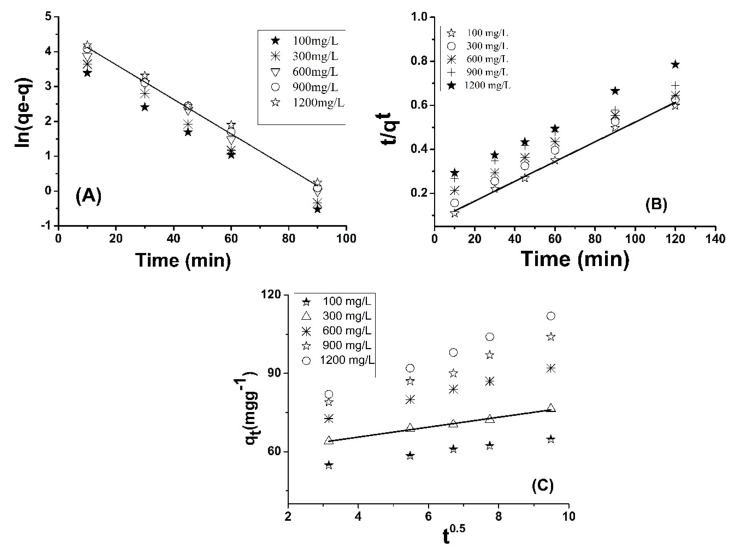
Pseudo first-order plot (**A**), pseudo second-order plot (**B**) and intra-particle diffusion plot (**C**) for Pb(II) adsorption onto chemically modified SM.

**Table 1 molecules-26-06885-t001:** Particle size, pore size, pore volume and surface area of bare and chemically modified silica monolith (SM).

	Average Particle Size (µm)	Pore Size (Å)	Pore Volume (cm^3^/g)	Surface Area (m^2^/g)
Bare SM	2.4	342	1.87	124
Chemically modified SM	2.8	310	1.74	113

**Table 2 molecules-26-06885-t002:** Adsorption isotherms parameters for Pb(II) adsorption on chemically modified SM.

Isotherm	Adsorption Isotherm Parameters	
Freundlich	1/*n*	0.334
	*K_F_* (mg/g)	7.431
	*R* ^2^	0.997
Langmuir	*q_max_* (mg/g)	574.71
	*K* (L/mg)	0.0603
	*R* ^2^	0.999
Temkin	*B_T_* (KJmol^−1^)	48.533
	*K_T_* (Lmg^−1^)	0.5541
	*R* ^2^	0.999

**Table 3 molecules-26-06885-t003:** Pseudo first-order, pseudo second-order and intra-particle diffusion values for Pb(II) adsorption onto chemically modified SM.

*C*_0_ (mg/L)	Pseudo First Order Experimental		Pseudo First Order Calculated		Pseudo Second Order Calculated			Intra-Particle Diffusion Values Calculated	
	*qe* (mg/g)	*K*1	*qe*	*R* ^2^	*qe* (mg/g)	*K* _2_	*R* ^2^	*R* ^2^	*K_i_* (mgg^−1^min^−1^)
100	235.30	0.0483	173.67	0.9993	165.34	0.0543	0.998	0.9943	3.35
300	340.62	0.0417	296.56	0.9926	322.12	0.0456	0.997	0.9978	2.34
600	412.80	0.0627	337.34	0.9978	387.30	0.0367	0.999	0.9945	4.56
900	441.31	0.0538	345.21	0.9998	392.71	0.0334	0.995	0.9825	4.56

**Table 4 molecules-26-06885-t004:** Comparison of adsorption capacity of the various adsorbents for Pb(II) removal.

Adsorbent	*qmax* (mg/g)	Ref
Cu–Mg Binary Ferrite	57.7	[54]
Hydrous manganese dioxide	140.3	[55]
Amino-grafted mesoporous silica (NH_2_-MCM-41)	54	[56]
Magnetic carbon nanotubes/diatomite	60	[57]
Chitosan/magnetite composite beads	63	[58]
Silica nano-powders/alginate	83	[59]
Magnetic hydroxypropyl CS/multiwalled carbon nanotubes	101	[60]
Nano-CS/sodium alginate/microcrystalline cellulose beads	114	[61]
Cotton stalk activated carbon	119	[62]
Poly(amidoxime)/SiO_2_	120	[63]
Bentonite-alginate composite	162	[64]
Mesoporous silica materials (MCM-48)	169	[65]
4-(chloro-2-mercaptophenyl) carbamodithioate (ACMPC) doped with mesoporous silica	188	[66]
Nanohydroxyapatite–alginate	236	[67]
Fumarate ferroxane	243	[68]
Cellulose nanocrystal/sodium alginate	338	[69]
Macroporous calcium alginate aerogel	390	[70]
Alg·S + SiO_2_	439	[71]
Alg·S + SiO_2_NH_2_	585	[71]
Hydrogel based on vinyl-functionalized alginate	784	[72]
CS/Fe-hydroxyapatite composite bead	1385	[51]
Chemically modified SM	792	This study

## Data Availability

Not applicable.

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
