# Peer review of "Efficient Removal of Pb(II) from Aqueous Medium Using Chemically Modified Silica Monolith"

_molecules, 2021, doi:10.3390/molecules26226885_

Round 1
Reviewer 1 Report
After reviewing the revised version of the manuscript prepared by Ashraf Ali, Sarah Alharthi, Imad Uddin, Javed Hussain, Fazal Mabood, Bashir Ahmad, Idrees Khan and Alia Naz entitled „Adsorptive removal of Pb(II) from aqueous medium using chemically modified silica monolith particles as efficient adsorbent” I find that most of my comments from the previous review have been incorporated in this version. The manuscript contains drawings from the previous version and new drawings, which makes it difficult to review. The authors did not correct the drawings illustrating the adsorption kinetics, they left the experimental points connected by a broken line in the coordinate system characteristic for the linear form of pseudo-first-order, pseudo-second-order and intra-partial diffusion models, Fig. 5. In my opinion, the line in Fig. 5 should show the result of matching individual models, which would complement, next to the R2 value in Table 2, the quality of the fit.
I believe that after correcting Fig. 5, the paper can be accepted for publication in the Molecules journal.
Author Response
Reviewer’s comment: The authors did not correct the drawings illustrating the adsorption kinetics, they left the experimental points connected by a broken line in the coordinate system characteristic for the linear form of pseudo-first-order, pseudo-second-order and intra-partial diffusion models, Fig. 5. In my opinion, the line in Fig. 5 should show the result of matching individual models, which would complement, next to the R2 value in Table 2, the quality of the fit.
I believe that after correcting Fig. 5, the paper can be accepted for publication in the Molecules journal.
Authors response: Adsorption kinetics data was re-computed and the drawings were corrected, the revised Fig 5 is added into the manuscript. We hope we have addressed the issue in Fig 5 which was raised by the respected reviewer.
Reviewer 2 Report
All my comments have been addressed then I recommend it accept.
Author Response
Reviewer's comment: All my comments have been addressed then I recommend it accept.
Authors response: Thank you so much for your time and valuable suggestions to improve the quality of our manuscript.
This manuscript is a resubmission of an earlier submission. The following is a list of the peer review reports and author responses from that submission.
Round 1
Reviewer 1 Report
Reviews for a paper focusing on the Adsorptive removal of Pb(II) from an aqueous medium using chemically modified silica monolith particles as efficient adsorbents. The paper is generally well-written, coherent, and technically sound. However, these are areas of improvement:
1. The topic could be revised to read-write. The proposed topic is: Efficient removal of Pb(II) from aqueous media using chemically modified silica monolith.
2. Quality control is missing in the data. The authors should consider emphasizing the quality control procedures and insert the standard deviations.
3. More modelling could be added to substantiate adsorption mechanisms. Mere Langmuir and Freundlich are not adequate to define mechanisms. Highlight chemisorption and all that.
4. Although type-setting will be done in the final stage. Using the same font and type will be appealing to the reviewers.
5. The comparison table should focus on materials of the same family. We compare apples to apples.
6. Figure captions should be linked to images. Image before and after adsorption should be created. EDS could further support the fate of chemical species.
Reviewer 2 Report
This work covers on an important topic of water pollution and the authors applied chemically modified silica monolith particles as efficient adsorbent to remove Pb(II) from aqueous medium, which is a good design and effective to address the environmental pollution. Porous silica monolith particles were prepared by sol-gel method and their surface modification of was carried out using trimethoxy silylpropyl urea (TSPU) to prepare inorganic-organic hybrid adsorbent. And the optimum conditions for adsorption of lead (Pb) were confirmed to be pH 3, adsorbent dose 0.4 g/L, Pb(II) ions concentration 500 mg/L and adsorption time 60 min. Besides, the adsorption of Pb(II) followed pseudo-second order model from their kinetic study. In whole, the research system of this paper is relatively complete. However, the part of characterization of related materials is not very professional and the description is not deep enough. Then I recommend major revision. Some comments are given as below:
(1) About the Morphology of the adsorbent (Sil-MSPU), the discussion is very confused for the readers. The scanning electron microscopy images of SMP and Sil-MSPU are shown in Fig 1. So what is the material in Figure 1A? The Author should mention it. “Figure 1. SEM images of SMP and Sil-MSPU adsorbent.” is confused. And the Figure 1A,1B,1C,1D should be defined.
(2) Similarly, the SEM images before and after the adsorption of lead (II) ions are shown in Fig 1(B & C) respectively. The magnification is inconsistent for the Figure 1B and 1C, so it is impossible to compare the surface roughness. The magnification should be the same.
(3) After the adsorption of lead (II) ions, so the authors should add the SEM mapping or EDX or XPS data to confirm the results and is there hybrid structure on the surface of Sil-MSPU?
(4) About the Figure arrangement, it is not reasonable. Some Figures should be combined together such as Figure 2. Effect of concentration on Pb(II) adsorption, Figure 3. Effect of adsorbent dose on Pb(II) adsorption, Fig 4 and Fig 5.
(5) Table 1. Adsorption isotherms parameters for Pb(II) adsorption on Sil-MSPU. 256. The form is not professional, and the bold font is not uniform.
(6) From the BET testing, how about the specific area of the related samples? If condition allowed, it should added the particles size influence on the performance of Pb(II) adsorption.
(7) Some recent adsorption materials (Rare Met., 2019; 38(1):73-80); carbon materials (Wang, DC., et al. A review of helical carbon materials structure, synthesis and applications; Rare Metals. 2021, 40, 3–19 and Rare Metals. 2021, 40, 1708-1718 and novel composite (Carbon 2021, 173, 185-193) should be cited, it may be used as the related adsorption to highlight this topic and materials design.
(8) Please check the whole paper to make the writing more professional without grammar errors.
Reviewer 3 Report
Manuscript presented to me for evaluation prepared by: Ashraf Ali, Sarah Alharthi, Imad Uddin, Javed Hussain, Fazal Mabood, Bashir Ahmad, Idrees Khan and Alia Naz entitled "Adsorptive removal of Pb (II) from aqueous medium using chemically modified silica monolith particles as efficient adsorbent ”describes the sol-gel method for obtaining modified silica trimethoxy silyl propyl urea, its characteristics by comparing SEM photos of modified and unmodified silica particles and the influence of Pb (II) ion adsorption on the surface morphology and the results of Pb (II) adsorption measurements. In my opinion, the characteristics of the samples are very poor (only SEM photos are included, and one of them differs in magnification), even though BET and TGA measurements were declared, the results of these measurements were not included. The characteristics of the tested samples lack the results of spectroscopic measurements that would allow the identification of functional groups on the silica surface.
Pb (II) ion adsorption measurements on the obtained silica samples were carried out by the uptake of Pb (II) ions from the solution, the concentration of Pb (II) ions was measured using the ASA spectrometry method. The following measurements were carried out:
- -effect of the initial concentration of Pb (II) ions on the removal of Pb (II) ions from the solution due to adsorption on Si-MSPU,
- Effect of adsorbent dose on the% removal of Pb (II) ions on Si-MSPU
- Measurements of the kinetics of Pb (II) adsorption on Si-MSPU
- Influence of pH on adsorption of Pb (II) ions on Si-MSPU
The presented manuscript is poorly written and has a number of shortcomings:
- The relationship presented in Fig. 2 suggests that the removal of Pb (II) ions with increasing concentration may be the result of the precipitation of Pb (II) ions from the solution. In the case of the adsorption process, the % ion removal from the solution decreases with increasing initial concentration. It is difficult to determine the adsorption conditions, as the authors report in the measurements that they used 1g (line 161) once and then 0.4 g (line 162) of Si-MSPU.
- The interpretation of the results presented in Fig. 5 is imprecise, the maximum adsorption occurs at pH ~ 4 and not at pH = 5.5 as given by the authors. The presented relationships show that the adsorbent has a very weak affinity for Pb (II) ions above pH ~ 4, which limits its practical application.
- The isotherms shown in Figures 6 and 7 apply to different concentration ranges. In the case of Fig. 6 showing the fit of the Freundlich isotherm from approx. 2.5 to 400, and in the case of the Langmuir isotherm from 0.2 to 6.5 (no units, presumably mg/dm3). The concentration range of the Fraundlich isotherm fit is narrow, unacceptable for the Langmuir isotherm (1 order concentration).
- In Fig. 8, 9, it was intended to present the kinetics of Pb (II) adsorption on Si-MSPU fitted with pseudo-first, pseudo-second and intra-particle diffusion models, while the experimental points were connected with broken lines, this is not a reflection of the fit with kinetic models.
- The manuscript does not present the results of Pb (II) desorption and adsorbent regeneration, there is only a brief change in the Conclusion.
- There is no comparison of Si-MSPU adsorption properties with silica.
Considering all the shortcomings, I propose not to accept the paper for publication in the Molecules journal.